# APP: Adaptive Prototypical Pseudo-Labeling for Few-shot OOD Detection

**Pei Wang**[1*], **Keqing He**[2*], **Yutao Mou**[1*], **Xiaoshuai Song**[1], **Yanan Wu**[1]
**Jingang Wang**[2], **Yunsen Xian**[2], **Xunliang Cai**[2], **Weiran Xu**[1*]

[1]Beijing University of Posts and Telecommunications, Beijing, China
[2]Meituan, Beijing, China

{wangpei,myt,songxiaoshuai,yanan.wu,xuweiran}@bupt.edu.cn
{hekeqing,wangjingang,xianyunsen,caixunliang}@meituan.com

## Abstract

Detecting out-of-domain (OOD) intents from user queries is essential for a task-oriented dialogue system. Previous OOD detection studies generally work on the assumption that plenty of labeled IND intents exist. In this paper, we focus on a more practical few-shot OOD setting where there are only a few labeled IND data and massive unlabeled mixed data that may belong to IND or OOD. The new scenario carries two key challenges: learning discriminative representations using limited IND data and leveraging unlabeled mixed data. Therefore, we propose an adaptive prototypical pseudo-labeling (APP) method for few-shot OOD detection, including a prototypical OOD detection framework (ProtoOOD) to facilitate low-resource OOD detection using limited IND data, and an adaptive pseudo-labeling method to produce high-quality pseudo OOD&IND labels. Extensive experiments and analysis demonstrate the effectiveness of our method for few-shot OOD detection.[1]

## 1 Introduction

Out-of-domain (OOD) intent detection learns whether a user query falls outside the range of pre-defined supported intents. It helps to reject abnormal queries and provide potential directions of future development in a task-oriented dialogue system (Akasaki and Kaji, 2017; Tulshan and Dhage, 2018; Lin and Xu, 2019; Xu et al., 2020; Zeng et al., 2021a,b; Wu et al., 2022a,b; Mou et al., 2022). Since OOD data is hard to label, we need to rely on labeled in-domain samples to facilitate detecting OOD intents.

Previous OOD detection studies generally work on the assumption that plenty of labeled IND intents exist. They require labeled in-domain data

to learn intent representations and then use scoring functions to estimate the confidence score of a test query belonging to OOD. For example, Hendrycks and Gimpel (2016) proposes Maximum Softmax Probability (MSP) to use maximum softmax probability as the confidence score and regard a query as OOD if the score is below a fixed threshold. Xu et al. (2020) further introduces another distance-based method, Gaussian discriminant analysis (GDA), which uses the maximum Mahalanobis distance (Mahalanobis, 1936) to all in-domain classes centroids as the confidence score. Although these models achieve satisfying performance, they all rely on sufficient labeled IND data, which limits their ability to practical scenarios.

In this paper, we focus on a more practical setting of OOD detection: there are only a few labeled IND data and massive unlabeled mixed data that may belong to IND or OOD. We call it as Few-Shot OOD Detection. Note that the mixed data don't have labeled IND or OOD annotations. Considering that unlabeled data is easily accessible, we believe this setting is more valuable to be explored. However, few-shot OOD detection carries two key challenges. (1) **Learning discriminative representations using limited IND data**: OOD detection requires discriminative intent representations to separate IND&IND intents and IND&OOD intents. However, traditional models based on cross-entropy or supervised contrastive learning (Zeng et al., 2021a) fail to distinguish intent types under the few-shot setting. Limited labeled IND data makes it hard to learn discriminative intent representations. (2) **Leveraging unlabeled mixed data**: Unlabeled data contains IND and OOD intents which both benefit in-domain intent recognition and OOD detection. But it's nontrivial to leverage the mixed data because we don't know prior knowledge of OOD data.

To solve the issues, we propose an **A**daptive **P**rototypical **P**seudo-labeling (APP) for Few-shot

---

*The first three authors contribute equally. Weiran Xu is the corresponding author.

[1]We release our code at https://github.com/Yupei-Wang/App-OOD.

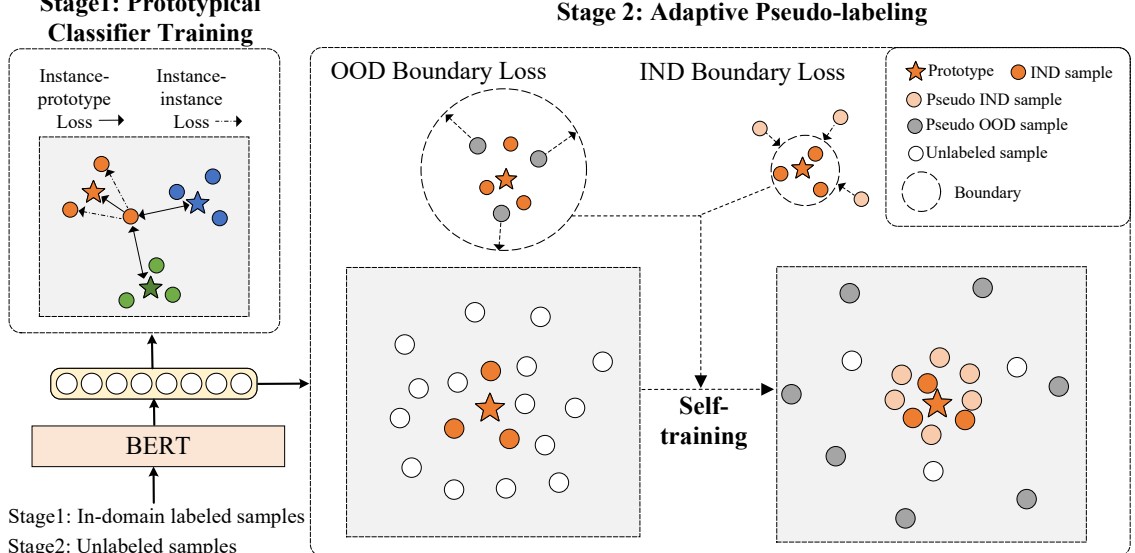

Figure 1: The overall architecture of our adaptive prototypical pseudo-labeling (APP) for few-shot OOD detection.

OOD Detection. To learn discriminative representations, we propose a prototypical OOD detection framework (ProtoOOD) using limited IND data. Inspired by the idea of PCL (Li et al., 2020), we introduce an instance-instance loss to pull together samples of the same class and an instance-prototype loss to enforce the prototypes to be center points of classes. After training a prototypical in-domain classifier using few-shot IND data, we compute the maximum cosine distance of an input query to all in-domain prototypes as the confidence score. If the score is above a fixed threshold, we believe it's an OOD intent. Compared to existing OOD detection methods (Hendrycks and Gimpel, 2016; Lin and Xu, 2019; Xu et al., 2020; Wu et al., 2022c; Mou et al., 2022), our prototypical OOD detection framework models rich class-level semantics expressed implicitly by training instances. Empirical experiments in Section 3.3 demonstrate our framework achieves superior performance both on IND and OOD metrics. To leverage unlabeled mixed data, we propose an adaptive pseudo-labeling method to iteratively label mixed data and update the prototypical IND classifier. We find typical pseudo-labeling methods (Lee, 2013; Cascante-Bonilla et al., 2020; Rizve et al., 2021) work poorly because the model can't produce high-quality pseudo IND&OOD labels, and get even worse performance. Therefore, we introduce two instance-prototype margin objectives to adaptively pull together pseudo IND samples and prototypes and push apart pseudo OOD samples. We aim to distinguish the confidence score distributions of

IND and OOD data by adjusting distances between IND or OOD samples and prototypes (see Section 4.2).

Our contributions are: (1) We propose an adaptive prototypical pseudo-labeling (APP) method for few-shot OOD detection. (2) We introduce a prototypical OOD detection framework (ProtoOOD) to learn discriminative representations and facilitate low-resource OOD detection using limited IND data, and an adaptive pseudo-labeling method to produce high-quality pseudo OOD&IND labels to leverage unlabeled mixed data. (3) Experiments and analysis demonstrate the effectiveness of our method for few-shot OOD detection.

## 2 Methodology

### 2.1 Problem Formulation

In the few-shot OOD detection setting, we assume that a limited labeled in-domain (IND) dataset $D_l = \{(\mathbf{x}_i, y_i)\}_{i=1}^n$ consists n samples drawn from IND, and an unlabeled dataset $D_u = \{(\mathbf{x}_i)\}_{i=1}^m$ consists m unlabeled samples drawn both ID and OOD. Note that we don't know whether an unlabeled sample in $D_u$ belongs to IND or OOD. Our goal is to distinguish whether an unknown test sample is drawn from ID or not using $D_l$ and $D_u$. Compared to traditional OOD detection setting, few-shot OOD detection carries two key challenges: learning discriminative representations using limited IND data and leveraging unlabeled mixed data.

## 2.2 Adaptive Prototypical Pseudo-Labeling

**Overall Architecture** Fig 1 shows the overall architecture of our proposed adaptive prototypical pseudo-labeling (APP) for few-shot OOD detection. APP includes two training stages. We first use the limited labeled IND data $D_l$ to train a prototypical in-domain classifier to learn discriminative intent representations. Then, we use an adaptive pseudo-labeling method to iteratively label mixed data $D_u$ and update the prototypical classifier. In the inference stage, we compute the maximum cosine similarity of an input test query to all in-domain prototypes as the confidence score. If the score is below a fixed threshold, we believe it's OOD.

**Prototypical OOD Detection** Previous OOD detection models (Hendrycks and Gimpel, 2016; Xu et al., 2020; Zeng et al., 2021a; Wu et al., 2022c; Mou et al., 2022) are customized for the setting with sufficient labeled IND data and lack generalization capability to few-shot OOD detection. We find these models suffer from the overconfidence issue (Liang et al., 2017a,b) where an OOD test sample even gets an abnormally high confidence score and is wrongly classified into in-domain types (see Section 4.1). Therefore, inspired by recent prototype learning work (Li et al., 2020; Cui et al., 2022), we propose a prototypical OOD detection framework (ProtoOOD) to learn discriminative intent representations in the few-shot setting.

As shown in Fig 1, we first get the hidden state of [CLS] to represent an input IND sample, then project it to another embedding space for prototype learning. The prototypes are used as class centroids. Denote $\mathcal{C} = \{\mathbf{c}_1, \cdots, \mathbf{c}_{|\mathcal{C}|}\}$ as the set of prototype vectors, which are randomly initialized. We introduce the following objectives. The first one is the instance-instance loss:

$$\mathcal{L}_{ins} = \sum_{i=1}^{N} -\frac{1}{N_{y_i} - 1} \sum_{j=1}^{N} \mathbf{1}_{i \neq j} \mathbf{1}_{y_i = y_j}$$
$$\log \frac{\exp(s_i \cdot s_j)}{\sum_{k=1}^{N} \mathbf{1}_{i \neq k} \exp(s_i \cdot s_k)} \quad (1)$$

where $s_i, s_j$ are projected features from [CLS] hidden states. $N_{y_i}$ is the total number of examples in the batch that have the same label as $y_i$ and $\mathbf{1}$ is an indicator function. This loss aims to pull together samples of the same class and push apart samples from different classes, which helps learn discriminative representations. The second is the instance-prototype loss:

$$\mathcal{L}_{\text{proto}} = \sum_{i=1}^{N} -\log \frac{\exp(s_i \cdot \mathbf{c}_i)}{\sum_{j=1}^{|\mathcal{C}|} (s_i \cdot \mathbf{c}_j)} \quad (2)$$

where $\mathbf{c}_i$ is the corresponding prototype of the sample $s_i$ and $|\mathcal{C}|$ the the total number of prototypes. This objective forces each prototype to lie at the center point of its instances. Our final training objective $\mathcal{L}_{pcl} = \mathcal{L}_{ins} + \mathcal{L}_{\text{proto}}$ combines the instance-instance loss and instance-prototype loss.

**Adaptive Pseudo-Labeling** Few-shot OOD detection has a large corpus of unlabeled data that contains mixed IND and OOD intents. How to exploit the data is vital to both benefiting in-domain intent recognition and OOD detection. However, it's nontrivial to apply existing semi-supervised methods (Lee, 2013; Cascante-Bonilla et al., 2020; Sohn et al., 2020) to leveraging the unlabeled data. Because prior knowledge of OOD data is unknown, making it hard to distinguish unlabeled IND and OOD intents simultaneously (see Section 4.2). Therefore, we propose an adaptive pseudo-labeling method to iteratively label mixed data and update the prototypical classifier.

Specifically, we design two thresholds $S$ and $L$ ($S < L$): if the maximum cosine similarity of an input query to all in-domain prototypes is higer than the larger threshold $L$, we believe it belongs to IND. And if the similarity is smaller than the smaller threshold $S$, we believe it belongs to OOD [2]. Then the pseudo IND samples can be directly used by optimizing $\mathcal{L}_{pcl}$. To use the pseudo OOD samples $x_i^{\text{OOD}}$, we propose an OOD instance-prototype margin objective to push away pseudo OOD samples from in-domain prototypes:

$$\mathcal{L}_{\text{ood}} = \max \left[0, \max_l \left(cos\left(x_i^{\text{OOD}}, \mathbf{c}_l\right) - \mathcal{M}_{OOD}\right)\right] \quad (3)$$

where $\mathbf{c}_l$ is the $l$-th prototype and $\mathcal{M}_{OOD}$ is a margin hyperparameter. However, we find simply applying $\mathcal{L}_{\text{ood}}$ can't produce correct pseudo IND samples. Because noisy pseudo OOD labels in $\mathcal{L}_{\text{ood}}$ may also have IND samples and will make the distances of all the samples including IND and OOD to prototypes larger. Therefore, we further propose an IND instance-prototype margin objec-

---

[2]In the implementation, we use the pre-trained prototype classifier in the first stage to compute the maximum cosine similarity of all the unlabeled data, and select the scores in the first 5th and last 5th positions after sorting in ascending order as the thresholds $S$ and $L$. These two thresholds are fixed in each pseudo-labeling process.

| Method | 5-shot | | | | | | 10-shot | | | | | | 20-shot | | | | | |
|---|---|---|---|---|---|---|---|---|---|---|---|---|---|---|---|---|---|---|
| | ALL | | IND | | OOD | | ALL | | IND | | OOD | | ALL | | IND | | OOD | |
| | ACC | F1 | ACC | F1 | Recall | F1 | ACC | F1 | ACC | F1 | Recall | F1 | ACC | F1 | ACC | F1 | Recall | F1 |
| LOF | 30.78 | 27.52 | 53.16 | 27.07 | 23.45 | 36.18 | 49.38 | 36.71 | 57.47 | 35.39 | 46.73 | 59.88 | 73.52 | 59.41 | **70.39** | 58.26 | 74.54 | 81.35 |
| GDA | 33.47 | 26.46 | 46.62 | 25.64 | 29.17 | 42.06 | 56.84 | 42.1 | 59.42 | 40.76 | 56.06 | 67.62 | 74.18 | 60.4 | 69.94 | 59.26 | 75.58 | 81.97 |
| UniNL | 40.64 | 27.51 | 44.25 | 26.17 | 39.45 | 53.03 | 61.68 | 43.67 | 58.63 | 42.14 | 62.67 | 72.72 | 76.43 | 60.02 | 70.39 | 58.74 | 78.41 | 83.9 |
| MSP | 55.66 | 37.99 | 53.51 | 36.42 | 56.37 | 67.74 | 62.37 | 46.34 | 61.45 | 44.95 | 62.7 | 72.73 | 73.52 | 59.41 | 70.39 | 58.26 | 74.54 | 81.35 |
| Energy | 58.55 | 39.57 | 53.55 | 37.92 | 60.19 | 70.97 | 66.66 | 48.87 | 60.24 | 47.38 | 68.76 | 77.19 | 77.80 | 62.52 | 68.16 | 61.32 | 80.94 | 85.06 |
| ProtoOOD(ours) | 70.23 | 51.14 | 60.97 | 49.64 | 73.26 | 79.83 | 75.27 | 58.75 | 66.89 | 57.47 | 78.02 | 83.1 | 79.98 | 63.61 | 67.63 | 62.38 | 84.03 | 87.1 |
| $\mathcal{L}_{pcl}$ | 63.21 | 48.44 | **65.13** | 47.15 | 62.59 | 72.95 | 74.55 | 59.87 | 68.42 | 58.68 | 76.55 | 82.57 | 75.39 | 61.88 | 67.89 | 60.77 | 77.84 | 83.03 |
| $\mathcal{L}_{pcl} + \mathcal{L}_{ind}$ | 59.22 | 43.05 | 60.92 | 41.65 | 58.66 | 69.69 | 70.44 | 54.50 | 69.11 | 55.24 | 68.88 | 77.31 | 75.78 | 59.87 | 68.55 | 58.64 | 78.15 | 83.43 |
| $\mathcal{L}_{pcl} + \mathcal{L}_{ood}$ | 54.71 | 55.74 | 56.71 | 45.12 | 54.05 | 57.45 | 68.57 | 54.67 | 59.61 | 53.38 | 71.51 | 79.26 | 83.96 | 67.53 | 67.24 | 66.35 | **89.44** | 89.92 |
| APP(ours) | **73.05** | **54.78** | 63.95 | **53.32** | **76.03** | **82.51** | **83.21** | **68.19** | **71.58** | **67.07** | **87.03** | **89.51** | **84.7** | **70.76** | 71.52 | **69.75** | 89.11 | **90.3** |

Table 1: Performance comparison on Banking. Here we report the results of 25% IND ratio under 5-shot, 10-shot and 20-shot. Results are averaged over three random runs.

tive to pull together pseudo IND samples $x_i^{\text{IND}}$ and prototypes:

$$\mathcal{L}_{\text{ind}} = \max \left[ 0, \max_l \left( \mathcal{M}_{IND} - cos \left( x_i^{\text{IND}}, \mathbf{c}_l \right) \right) \right]$$
(4)

where $\mathcal{M}_{IND}$ is a margin hyperparameter [3] and $cos$ is the cosine similarity. We show the number of correct predicted pseudo-labeled IND and OOD samples in Fig 6 and find $\mathcal{L}_{\text{ood}}$ and $\mathcal{L}_{\text{ind}}$ systematically work as a whole and reciprocate each other. We aim to distinguish the confidence score distributions of IND and OOD data by adjusting distances between IND or OOD samples and prototypes. The final training loss in the second stage is $\mathcal{L} = \mathcal{L}_{pcl} + 0.05\mathcal{L}_{ind} + 0.05\mathcal{L}_{ood}$. We show he overall training process in Figure 1 and summarize the pseudo-code of our method in Algorithm 1.

## 3 Experiments

### 3.1 Datasets

We evaluate our method on two commonly used OOD intent detection datasets, Banking (Casanueva et al., 2020) and Stackoverflow (Xu et al., 2015). Following previous work (Mou et al., 2022), we randomly sample 25%, 50%, and 75% intents as the IND intents, and regard all remaining intents as OOD intents. To verify the effectiveness of our method on few-shot OOD detection with mixed unlabeled data, we divide the original training sets of the two datasets into two parts. We randomly sample $k = 5, 10, 20$ instances in each IND intent from the original training set to construct few-shot labeled IND dataset, while the remaining training samples are treated as the unlabeled mixed data including both IND and OOD class. We only

---

**Algorithm 1** : Adaptive Prototypical Pseudo-Labeling

**Require:** training dataset $\mathbf{D}_L = \{(x_i, y_i)\}_{i=1}^n$ and $\mathbf{D}_U = \{(x_i^{OOD})\}_{i=1}^m$, training steps $S$, epoch $E$

**Ensure:** a OOD intent detection model, which can classify an input query to either one IND class or OOD class. $\mathcal{Y} = \{1, \ldots, N\} \cup \{OOD\}$.

1: randomly initialize prototype embedding $\mu_j, j = 1, 2, ..., N$.
2: **for** step = 1 to $S$ **do**
3:     sample a mini-batch $\mathbf{B}$ from $\mathbf{D}_L$
4:     get the embedding $z_i$ of sample $x_i$ through the projection layer
5:     compute $\mathcal{L}_{ins}$ and $\mathcal{L}_{proto}$ ▷ **prototypical contrastive representation learning**
6:     update the network parameters and prototype vectors
7: **end for**
8: **for** epoch = 1 to $E$ **do**
9:     align sample $x_i$ from $\mathbf{D}_U$ with prototypes $\mu_j$
10:     compute $\mathcal{L}_{ins}$, $\mathcal{L}_{proto}$ and $\mathcal{L}_{IND}$ for pseudo-IND samples
11:     compute $\mathcal{L}_{OOD}$ for pseudo-OOD samples
12:     add $\mathcal{L}_{ins}$, $\mathcal{L}_{proto}$, $\mathcal{L}_{IND}$ and $\mathcal{L}_{OOD}$ together, and jointly optimize them ▷ **prototypical contrastive representation learning and adaptive margin learning**
13:     update the network parameters and prototype vectors.
14: **end for**

---

[3]We leave out these hyperparameters to Implementation Details in Section A.3.

use the few-shot labeled IND dataset to train the prototypical in-domain classifier. And we adopt the adaptive pseudo-labeling method to label unlabeled mixed IND and OOD data. Besides, since we focus on few-shot OOD detection setting, we also reduce the number of IND samples in the development set. Test set is the same as orginal data. More detailed information of dataset is shown in Appendix A.1.

### 3.2 Baselines

To verify the effectiveness of our prototypical OOD Detection (ProtoOOD) method under the few-shot OOD detection setting. We compare ProtoOOD with OOD detetcion baselines using recent baselines. For the feature extractor, we use the same BERT (Devlin et al., 2019) as backbone. We compare our method with the training objective cross-entropy(CE) and the scoring functions including MSP (Hendrycks and Gimpel, 2017), LOF (Lin and Xu, 2019), GDA (Xu et al., 2020) and Energy (Wu et al., 2022c). Besides, we also compare our method with the state-of-the-art baselines UniNL(Mou et al., 2022). We supplement the details of relevant baselines in the appendix A.2.

For the self-training on unlabeled mixed data, we compare our adaptive prototypical pseudo-labeling (APP) method with three different training objectives as the baselines of our unlabeled data training. (1) $\mathcal{L}_{pcl}$: For pseudo IND samples, we only use instance-instance loss and instance-prototype loss for training. (2) $\mathcal{L}_{pcl} + \mathcal{L}_{ind}$: For pseudo IND samples, in addition to instance-instance loss and instance-prototype loss training, the IND margin objective is added. (3) $\mathcal{L}_{pcl} + \mathcal{L}_{ood}$: For pseudo IND samples, we use instance-instance loss and instance-prototype loss, and OOD margin objective training for pseudo OOD samples.

### 3.3 Main Results

Table 1 and 2 show the performance comparison of different methods on Banking and Stackoverflow dataset respectively. In general, our proposed prototypical OOD detection framework (ProtoOOD) and adaptive prototypical pseudo-labeling method (APP) consistently outperform all the baselines with a large margin. Next, we analyze the results from two aspects:

(1) **Advantage of prototypical OOD detection.** We compare our proposed ProtoOOD with previous OOD detection baselines. Experimental results show that ProtoOOD is superior to other methods in both IND and OOD metrics, and the less labeled

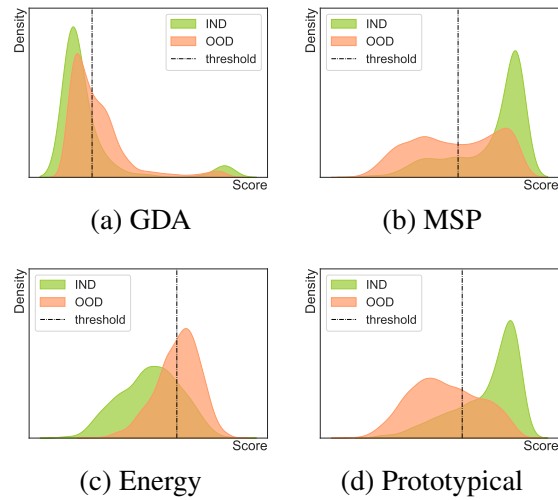

Figure 2: Score distribution curves of IND and OOD data using different scoring functions.

IND data, the more obvious the advantages of it. For example, on Banking dataset, ProtoOOD outperforms previous state-of-the-art baseline Energy by 2.04%(OOD F1), 2.18%(ALL ACC) on 20-shot setting, 5.91%(OOD F1), 8.61%(ALL ACC) on 10-shot setting and 8.89%(OOD F1), 11.68%(ALL ACC) on 5-shot setting. We also observed that the performance of previous methods under the 5-shot setting decreased significantly. We think that this is because previous methods rely on a large number of labeled IND samples to learn the generalized intent representations. LOF and KNN-based detection methods are easily affected by outliers. GDA will be affected by the inaccurate estimation of the covariance of the IND cluster distribution under the few-shot setting. MSP and Energy will encounter overconfidence problems due to the over-fitting of the neural network under the few-shot setting, resulting in the wrong detection of OOD as IND. In contrast, ProtoOOD is insensitive to outlier samples and has good generalization ability.

(2) **Comparison of different self-training methods.** We introduce two instance-prototype margin objectives for prototype-based self-training on unlabeled data. To understand the importance of the two margin objectives on our adaptive prototypical pseudo-labeling method, we perform an ablation study. The experimental results show that the joint optimization of $\mathcal{L}_{pcl}$, $\mathcal{L}_{ind}$ and $\mathcal{L}_{ood}$ achieves the best performance. We think that since there are not only IND samples but also a large number of OOD samples in the unlabeled data, it is challenging to directly use the model pre-trained on

| Method | 5-shot | | | | | | 10-shot | | | | | | 20-shot | | | | | |
|---|---|---|---|---|---|---|---|---|---|---|---|---|---|---|---|---|---|---|
| | ALL | | IND | | OOD | | ALL | | IND | | OOD | | ALL | | IND | | OOD | |
| | ACC | F1 | ACC | F1 | Recall | F1 | ACC | F1 | ACC | F1 | Recall | F1 | ACC | F1 | ACC | F1 | Recall | F1 |
| LOF | 22.8 | 18.37 | 34.47 | 15.99 | 18.91 | 30.29 | 29.14 | 28.52 | 54.50 | 27.75 | 20.69 | 32.32 | **79.18** | 46.51 | 26.13 | 38.31 | 96.87 | 87.49 |
| GDA | 28.03 | 24.56 | 39.47 | 22.09 | 24.22 | 36.92 | 35.04 | 27.66 | 51.74 | 24.66 | 29.48 | 42.66 | 56.57 | 48.5 | 70.27 | 45.19 | 52.00 | 65.05 |
| UniNL | 32.05 | 20.55 | 26.53 | 15.24 | 33.80 | 47.09 | 39.44 | 34.66 | 51.60 | 31.76 | 35.39 | 49.16 | 61.58 | 51.59 | 68.3 | 47.76 | 59.34 | 70.69 |
| MSP | 36.45 | 27.32 | 41.2 | 23.1 | 34.87 | 48.42 | 42.78 | 36.94 | 55.47 | 33.83 | 38.56 | 52.47 | 58.08 | 49.27 | 69.27 | 45.77 | 54.36 | 66.81 |
| Energy | 37.63 | 27.38 | 44.07 | 22.95 | 35.49 | 49.52 | 45.75 | 39.44 | 57.53 | 36.20 | 41.82 | 55.64 | 62.3 | 51.5 | 67.73 | 47.5 | 60.49 | 71.45 |
| ProtoOOD(ours) | 41.75 | 34.47 | 53.47 | 31.02 | 37.84 | 51.69 | **49.22** | **43.07** | 63.16 | 39.89 | 44.58 | 58.48 | 62.63 | 52.06 | 67.87 | 48.11 | **60.89** | **71.79** |
| $\mathcal{L}_{pcl}$ | 36.95 | 34.84 | 45.27 | 32.29 | 34.18 | 47.59 | 51.27 | 46.36 | 58.13 | 43.33 | 48.98 | 61.5 | 67.78 | 55.95 | 61.02 | 51.4 | 70.04 | 78.67 |
| $\mathcal{L}_{pcl} + \mathcal{L}_{ind}$ | 35.62 | 33.93 | 49.01 | 31.80 | 31.16 | 44.56 | 47.40 | 45.33 | 54.05 | 42.76 | 45.2 | 58.17 | 58.05 | 55.55 | **69.2** | 53.33 | 54.33 | 66.65 |
| $\mathcal{L}_{pcl} + \mathcal{L}_{ood}$ | 54.82 | 40.24 | **54.14** | 34.79 | 55.09 | 67.53 | 72.08 | 59.30 | 61.47 | 54.92 | 75.62 | 81.25 | 83.3 | 62.72 | 48.67 | 57.32 | **94.84** | 89.72 |
| APP(ours) | **58.35** | **44.87** | 51.53 | **39.68** | **60.62** | **70.81** | **75.22** | **65.06** | 61.67 | **61.24** | **79.73** | **84.18** | **85.17** | **73.45** | 65.47 | **70.02** | 91.73 | **90.61** |

Table 2: Performance comparison on Stackoverflow. Here we report the results of 25% IND ratio under 5-shot, 10-shot and 20-shot. Results are averaged over three random runs.

few-shot labeled IND samples for pseudo-labeling, which will limit the performance of prototype-based self-training. After introducing two instance-prototype margin objectives, which adaptively adjust the distance between the IND/OOD samples and prototypes, we can obtain more reliable pseudo labels stably. We also find that when the two margin objectives are used alone, they will hinder the prototype-based self-training. This shows that when we use instance-prototype margin objectives for prototype-based self-training, we need to constrain the distance from IND and OOD to the prototypes at the same time. Besides, in order to explore the adaptability of adaptive prototypical pseudo-labeling on other OOD detection methods, we discuss it in Appendix C. The conclusion is that our APP method has strong generality and is compatible with other OOD detection methods well.

## 4 Qualitative Analysis

### 4.1 Effect of Prototypical OOD Detection

In few-shot OOD detection, how to use limited IND data to learn discriminative intent representations is a key challenge. In order to compare the performance of different representation learning objectives under few-shot settings, we perform intent visualization of CE, KNCL (Mou et al., 2022) and our prototypical contrastive learning objective, as shown in Fig 3. The results show that the prototype-based method can learn more compact IND clusters, and the distance between OOD and IND is farther, which facilitates OOD detection.

To analyze the performance of different scoring functions in the few-shot setting, we compare the GDA, MSP, Energy and our proposed Prototype-based score distribution curves for IND and OOD data, as shown in Fig 2. The smaller the overlap-ping area of IND and OOD intents means that it is more beneficial to distinguish IND and OOD. We can see that the prototypical score can better distinguish IND and OOD under the few-shot setting, while other scores suffer serious IND and OOD overlapping. This also shows that compared with the previous OOD detection methods, our ProtoOOD method can learn more generalized representations under the few-shot setting, which is beneficial to distinguish IND and OOD.

### 4.2 Effect of Adaptive Pseudo-Labeling

**Change of IND and OOD score distribution** In order to further explore the advantages of adaptive pseudo-Labeling (APP), we show the change of IND and OOD score distribution curves during the self-training process in Fig 4. We can clearly see that as the training process goes on, IND and OOD intents are gradually separated, and there is no case that the OOD samples are too close to the prototypes or the IND samples are too far away from the prototypes. We also show the change of score distribution curves of other self-training variants in Appendix A.2. It can be seen that when we remove $\mathcal{L}_{ood}$, a large number of OOD samples will be identified as IND; When we remove $\mathcal{L}_{ind}$, a large number of IND samples will be classified as OOD. We think that this is because the two instance-prototype margin objectives constrain the distance between IND/OOD and class prototypes respectively. It is necessary to constrain the distances between IND and OOD to the class prototypes at the same time. This also shows that the two instance-prototype margin objectives is beneficial to obtain reliable pseudo labels stably.

**Changes of class prototypes** In addition, we also observe the changes of class prototypes during adaptive pseudo-labeling in Fig 5. We can

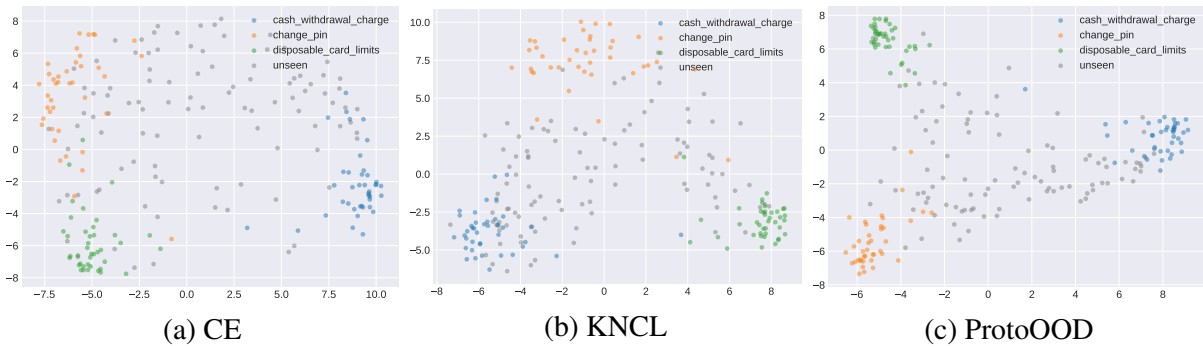

(a) CE  (b) KNCL  (c) ProtoOOD

Figure 3: Visualization of IND and OOD intents using different IND pre-training losses.

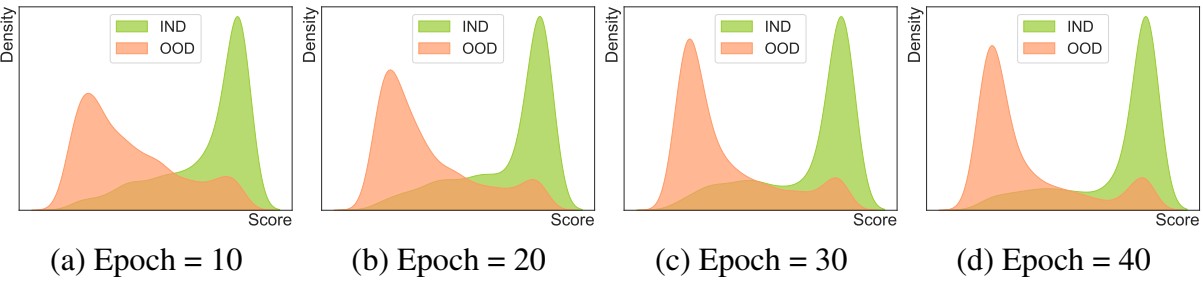

(a) Epoch = 10  (b) Epoch = 20  (c) Epoch = 30  (d) Epoch = 40

Figure 4: Change of score distribution curves of IND and OOD data during adaptive pseudo-labeling.

see that the class prototypes are gradually close to the center of IND clusters, and away from OOD data. This intuitively reflects that our APP method can adaptively adjust the distance between IND or OOD samples and class prototypes by adding two instance-prototype margin objectives, which facilitates pseudo-labeling.

**Changes of the number of correct pseudo labels** We also count the number of correct pseudo labels in the self-training process. Fig 6 shows the results. The horizontal axis is the epoch of self-training, and the vertical axis is the correct number of pseudo labels. It can be seen that when only using $\mathcal{L}_{pcl}$, more and more IND data can be correctly detected, but OOD data cannot be effectively detected. We believe that this is because we only pre-train the model on few-shot labeled IND data, and did not encounter OOD samples during the model pre-training, so simply using $\mathcal{L}_{pcl}$ for self-training cannot get reliable OOD pseudo labels. When we use $\mathcal{L}_{pcl} + \mathcal{L}_{ood}$, we get the opposite result, we can only detect more and more OOD samples. We think that it is because the OOD margin objective only pushes the prototype far away from OOD data, but it cannot pull it close to more IND data, resulting in the model can not correctly detect more IND data. In contrast, when we use three objectives for joint optimization, we get more

| Threshold | ALL | | IND | | OOD | |
|---|---|---|---|---|---|---|
| | ACC | F1 | ACC | F1 | Recall | F1 |
| $\mathcal{L}_{pcl} + \mathcal{L}_{ood}$ | 72.08 | 59.30 | 61.47 | 54.92 | 75.62 | 81.25 |
| T=1 | **75.32** | 60.94 | 61.67 | 56.38 | **79.87** | 83.72 |
| T=5 | 75.22 | **65.06** | **61.67** | **61.24** | 79.73 | **84.18** |
| T=10 | 72.75 | 60.71 | 56.87 | 56.35 | 78.04 | 82.47 |

Table 3: Effect of threshold on the training performance of adaptive pseudo-labeling on stackhoverflow 25% IND ratio under 10-shot. Here we compare the results with $\mathcal{L}_{pcl} + \mathcal{L}_{ood}$ to prove the robustness of APP.

and more correct pseudo IND samples and pseudo OOD samples. This is why APP achieves the best effect compared with the other two self-training methods. It can promote prototype optimization towards the direction of being close to IND and far from OOD. Since using $\mathcal{L}_{pcl} + \mathcal{L}_{ind}$ gets the similar results as using only $\mathcal{L}_{pcl}$, so we only show the results of $\mathcal{L}_{pcl}$ for brevity.

### 4.3 Hyper-parameter Analysis

**The effect of the threshold for pseudo-labeling.** We use $T$ to represent the position of the threshold selected when pseudo-labeling. For example, $T=5$ means that we choose the fifth highest score as the upper threshold $L$ and the fifth lowest score as the lower threshold $S$. We compare the effect of different $T$ on the OOD detection performance as shown in Table 3. It can be seen that when

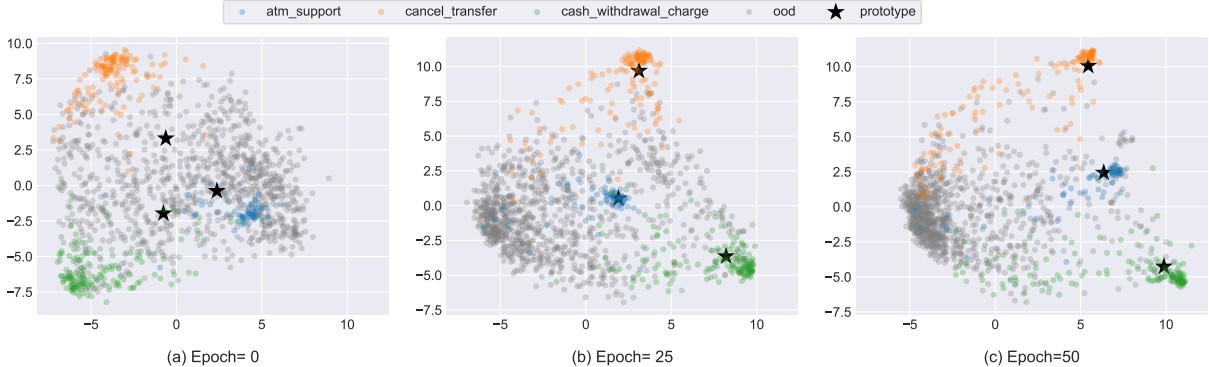

Figure 5: Visualization of IND and OOD intents during adaptive pseudo-labeling.

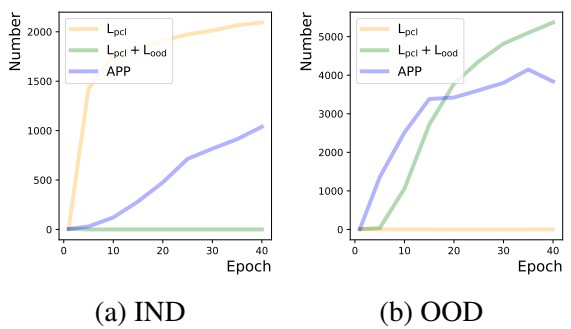

(a) IND      (b) OOD

Figure 6: Numbers of correct pseudo-labeled IND and OOD samples.

| Model | ALL | | IND | | OOD | |
|---|---|---|---|---|---|---|
| | ACC | F1 | ACC | F1 | Recall | F1 |
| $\mathcal{L} = \mathcal{L}_{pcl} + 0.03\mathcal{L}_{ind} + 0.03\mathcal{L}_{ood}$ | 81.10 | 66.06 | **72.11** | 66.21 | 85.03 | 88.79 |
| $\mathcal{L} = \mathcal{L}_{pcl} + 0.05\mathcal{L}_{ind} + 0.05\mathcal{L}_{ood}$ | **83.21** | **68.19** | 71.58 | **67.07** | **87.03** | 89.51 |
| $\mathcal{L} = \mathcal{L}_{pcl} + 0.07\mathcal{L}_{ind} + 0.07\mathcal{L}_{ood}$ | 82.83 | 67.09 | 70.89 | 65.97 | 86.70 | **90.23** |

Table 4: Effect of cofficients on the training performance of adaptive pseudo-labeling on banking 25% IND ratio under 10-shot.

$T$ takes a smaller value, it can often bring better OOD detection performance. This is because the stricter threshold brings higher accuracy of pseudo-labeling, thus improving the detection performance. Moreover, under different T, our method APP all exceeds the performance of other methods, which proves APP is robust for different T.

**The effect of the Margin Value for IND and OOD margin loss.** We select six different sets of IND margin and OOD margin values to illustrate the impact of Margin Values on detection performance as shown in Table 5. It can be seen that margin value has little effect on the OOD detection performance, which illustrates the robustness of our proposed margin loss.

**The effect of coefficients for IND and OOD margin loss.** We select three different coefficient combination of loss to illustrate the impact of co-

| Margin | ALL | | IND | | OOD | |
|---|---|---|---|---|---|---|
| | ACC | F1 | ACC | F1 | Recall | F1 |
| $\mathcal{L}_{pcl} + \mathcal{L}_{ood}$ | 72.08 | 59.30 | 61.47 | 54.92 | 75.62 | 81.25 |
| $\mathcal{M}_{IND} = 3.0 \; \mathcal{M}_{OOD} = 1.1$ | 72.5 | 60.41 | 56.32 | 56.54 | 78.14 | 82.27 |
| $\mathcal{M}_{IND} = 2.9 \; \mathcal{M}_{OOD} = 1.1$ | 72.40 | 61.08 | **61.87** | 56.91 | 75.91 | 81.94 |
| $\mathcal{M}_{IND} = 2.8 \; \mathcal{M}_{OOD} = 1.1$ | 72.75 | 60.71 | 56.87 | 56.35 | 78.04 | 82.47 |
| $\mathcal{M}_{IND} = 2.8 \; \mathcal{M}_{OOD} = 1.2$ | 72.97 | 60.73 | 56.07 | 56.32 | **78.60** | 82.80 |
| $\mathcal{M}_{IND} = 2.8 \; \mathcal{M}_{OOD} = 1.1$ | 72.75 | 60.71 | 56.87 | 56.35 | 78.04 | 82.47 |
| $\mathcal{M}_{IND} = 2.8 \; \mathcal{M}_{OOD} = 1.0$ | **73.32** | **63.67** | 59.47 | **59.82** | 77.93 | **82.90** |

Table 5: Effect of Margin Values on the training performance of adaptive pseudo-labeling on StackOverflow 25% IND ratio under 10-shot. Here we compare the results with $\mathcal{L}_{pcl} + \mathcal{L}_{ood}$ to prove the robustness of APP.

efficients. As shown in Table 5, it can be seen that coefficient has little effect on the OOD detection performance, which illustrates the robustness of our method.

### 4.4 Effect of Different Ratios of OOD Data

We compared the effect of different ratios of OOD Data. The results are shown in Table 9. We can see our proposed prototypical OOD Detection outperforms Energy on all IND ratios. It shows the effectiveness of protoOOD in few-shot OOD detection. After the self-training of unlabeled data, the effect of both IND classification and OOD detection are improved. As the ratio of OOD decreases, we find that F1-OOD is decreasing and F1-IND is increasing. some OOD samples are more likely to be confused with one of the IND intents. The number of IND classes increases the prior knowledge available for IND learning, enabling the model to learn better representations of IND and distinguish it from OOD.

### 4.5 Few-shot OOD detection of ChatGPT

Language Learning Models (LLMs) such as Chat-GPT [4] have demonstrated strong capabilities across a variety of tasks. In order to compare the perfor-

---

[4]https://openai.com/blog/ChatGPT

| Model | ALL | | IND | | OOD | |
|---|---|---|---|---|---|---|
| | ACC | F1 | ACC | F1 | Recall | F1 |
| Chatgpt (0-shot) | 51.4 | 45.8 | 77.94 | 45.11 | 42.72 | 58.84 |
| Chatgpt (1-shot) | 48.96 | 47.11 | 82.46 | 46.72 | 38.26 | 54.45 |
| Chatgpt (3-shot) | 47.68 | **49.56** | 86.2 | 49.47 | 35.03 | 51.35 |
| Chatgpt (5-shot) | 44.39 | 47.11 | **89.78** | 47.21 | 29.57 | 45.32 |
| APP(5-shot) | 58.35 | 44.87 | 51.53 | 39.68 | 60.62 | 70.81 |

Table 6: LLM's OOD detection ability on banking 50IND ratio.

mance of ChatGPT in OOD detection with our method, we conduct comparative experiments.

The prompt we use is: <Task description> You are an out-of-domain intent detector, and your task is to detect whether the intents of users' queries belong to the intents supported by the system. If they do, return the corresponding intent label, otherwise return unknown. The supported intents include: [Intent 1] ([Example1] [Example 2]...), ... The text in parentheses is the example of the corresponding intent. <Response format> Please respond to me with the format of "Intent: XX" or "Intent: unknown". <Utterance for test> Please tell me the intent of this text: [Here is the utterance for text.]

Results are shown in Figure 6. It shows that ChatGPT's effectiveness in identifying OOD samples is notably low, as shown by the significantly low Recall-OOD of 29.57 for 5-shot. Through case studies, we discover that it often misclassifies OOD samples as IND. This might be due to a clash between its broad general knowledge and specific domain knowledge. It further suggests that **it struggles to filter out unusual samples**, which is usually the first step in practical uses of the OOD task and constitutes the main goal of this task. On a positive note, its broad general knowledge has given it strong abilities to classify IND samples. We hope to explore strategies to merge the strengths of both models to improve overall performance in our future work.

## 5 Related Work

**OOD Detection** Previous OOD detection works can be generally classified into two types: supervised (Fei and Liu, 2016; Kim and Kim, 2018; Larson et al., 2019; Zheng et al., 2020) and unsupervised (Bendale and Boult, 2016; Hendrycks and Gimpel, 2017; Shu et al., 2017; Lee et al., 2018; Ren et al., 2019; Lin and Xu, 2019; Xu et al., 2020) OOD detection. The former indicates that there are extensive labeled OOD samples in the training data. Fei and Liu (2016); Larson et al. (2019), form a *(N+1)*-class classification problem

where the *(N+1)*-th class represents the OOD intents. We focus on the unsupervised OOD detection setting where labeled OOD samples are not available for training. Unsupervised OOD detection first learns discriminative representations only using labeled IND data and then employs scoring functions, such as Maximum Softmax Probability (MSP) (Hendrycks and Gimpel, 2017), Local Outlier Factor (LOF) (Lin and Xu, 2019), Gaussian Discriminant Analysis (GDA) (Xu et al., 2020), Energy (Wu et al., 2022c) to estimate the confidence score of a test query. Inspired by recent prototype models (Li et al., 2020; Cui et al., 2022) for few-shot learning, we propose a prototypical OOD detection framework (ProtoOOD) to facilitate low-resource OOD detection using limited IND data. Besides, (Zhan et al., 2022) is about few-shot OOD detection utilizing generation model to create more IND and OOD.

**Self-Supervised Learning** is an active research area, including pseudo-labeling (Lee, 2013; Cascante-Bonilla et al., 2020), consistency regularization (Verma et al., 2019; Sohn et al., 2020) and calibration (Yu et al., 2019; Xia et al., 2018). Since we focus on the few-shot OOD detection, we use the simple pseudo-labeling method and leave other methods to future work. Different from existing self-supervised learning work based on the assumption that all the data is drawn from the same distribution, few-shot OOD detection faces the challenge of lack of prior OOD knowledge, making it hard to distinguish unlabeled IND and OOD intents simultaneously. Therefore, we propose an adaptive pseudo-labeling method to iteratively label mixed data and update the prototypical classifier.

## 6 Conclusion

In this paper, we establish a practical OOD detection scenario: there are only a few labeled IND data and massive unlabeled mixed data that may belong to IND or OOD. We find existing OOD work can't effectively recognize OOD queries using limited IND data. Therefore, we propose an adaptive prototypical pseudo-labeling (APP) method for few-shot OOD detection. Two key components are the prototypical OOD detection framework (ProtoOOD) and adaptive pseudo-labeling strategy. We perform comprehensive experiments and analysis to show the effectiveness of APP. We hope to provide new insight of OOD detection and explore more self-supervised learning methods for future work.

## Limitations

In this paper, we propose an adaptive prototypical pseudo-labeling (APP) method for few-shot OOD detection, including a prototypical OOD detection framework (ProtoOOD) and adaptive pseudo-labeling method. Although our model achieves excellent performance, some directions are still to be improved. (1) We consider a basic self-supervised learning (SSL) method, pseudo-labeling. More other SSL methods should be considered. (2) Although our model achieves superior performance than the baselines, there is still a large gap to be improved compared to the full-data OOD detection. (3) Apart from SSL, unsupervised representation learning methods (Li and Xiangling, 2022; Zeng et al., 2021c) also make an effect, which are orthogonal to our pseudo-labeling method.

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

## A  Experiment Setups

### A.1  Dataset

Since the setting we focus on is few-shot OOD detection with massive mixed unlabeled data, we adjust the original data set. Table 7 shows the statistical information of the adjusted dataset of 25% IND ratio under 10-shot. Taking Stackoverflow as an example, it contains 20 intents. First, we divide all intents, randomly select 5 intents as IND intents, and the other 15 intents as OOD intents. Then we randomly select 10 instances for each IND intent as labeled IND training data to participate in

| Statistic | Banking | Stackoverflow |
|---|---|---|
| Avg utterance length | 12 | 10 |
| Intents | 77 | 20 |
| IND Intents | 19 | 5 |
| OOD Intents | 58 | 15 |
| Labeled training set size | 190 | 50 |
| Labeled training samples per class | 10 | 10 |
| Unlabeled Mixed training set size | 8813 | 11950 |
| Unlabeled IND training samples | 2092 | 2950 |
| Unlabeled OOD training samples | 6721 | 9000 |
| Unlabeled IND training samples per class | - | 590 |
| Unlabeled OOD training samples per class | - | 600 |
| Development set size | 190 | 50 |
| Development samples per class | 10 | 10 |
| Testing Set Size | 3080 | 6000 |
| Testing samples per class | - | 300 |

Table 7: Statistics of the few-shot OOD detection datasets of 25% IND ratio under 10-shot.

| Method | ALL | | IND | | OOD | |
|---|---|---|---|---|---|---|
| | ACC | F1 | ACC | F1 | Recall | F1 |
| SCL+GDA | 68.38 | 49.29 | 57.37 | 47.72 | 71.98 | 79.02 |
| protoOOD(ours) | **75.27** | **58.75** | **66.89** | **57.47** | **78.02** | **83.1** |
| Pseudo-labeling(SCL+GDA) | 73.6 | 56.25 | 67.11 | 54.91 | 75.73 | 81.64 |
| APP(ours) | **83.21** | **68.19** | **71.58** | **67.07** | **87.03** | **89.51** |

Table 8: Performance comparison on Banking 25% IND ratio. We report the performance of SCL+GDA and our prototypical OOD detection under 10-shot and the performance of the two methods after using the same pseudo-labeling strategy.

the few-shot training in the first stage. Therefore, each intent has 10 labeled training samples, and we have a total of 50 labeled training data. The remaining data in the original training set is treated as unlabeled data to participate in the second stage self-training. Therefore, there are 590 unlabeled samples left for each IND intent and 600 unlabeled samples left for OOD intent. A total of 11950 pieces of data is the training set of the second self-training stage, and their labels are unknown during training. Besides, we only retain 10 samples for each IND intent in the development set and the test set is consistent with the original dataset.

### A.2 Baselines

**MSP** (Maximum Softmax Probability)(Hendrycks and Gimpel, 2017) uses maximum softmax probability as the confidence score. If the score is lower than a fixed threshold, the query is regarded as OOD. MSP can easily lead to overconfidence in OOD in low-resource scenarios.

**LOF** (Local Outlier Factor)(Lin and Xu, 2019) uses the local outlier factor to detect unknown intents. It detects OOD by comparing the local density of a test query with its k-nearest neighbor's. If a query's local density is significantly lower than its k-nearest neighbor's, it is more likely to be regarded as OOD. LOF needs to check the local density of the sample, which is more vulnerable to the interference of few-shot, leading to the result that is not robust.

**GDA** (Gaussian Discriminant Analysis)(Xu et al., 2020) is a generative distance-based classifier to detect OOD samples. It estimates the class-conditional distribution on feature spaces of DNNs via Gaussian discriminant analysis, and then applies Mahalanobis distance to measure the confi-

dence score. However, the covariance matrix required to calculate Mahalanobis distance is difficult to accurately estimate with limited ID samples, which will seriously affect the performance of OOD detection.

**Energy** (Wu et al., 2022c) maps a sample x to a single scalar called the energy. It uses the threshold on the energy score to consider whether a test query belongs to OOD.

**UniNL**(Mou et al., 2022) uses a unified neighborhood learning framework to detect OOD intents, in which a KNCL objective is employed for IND pre-training and a KNN-based score function is used for OOD detection.Because KNN detection needs to calculate the distance between the test sample and the training sample, the limited training set makes the KNN detection result easy to be disturbed by abnormal points.

### A.3 Implementation Details

For the training of prototypical OOD detection, we use BERT to embed tokens. We initialize the prototype embeddings randomly with dimension of 256 and optimize the loss function with Adam optimizer (Kingma and Ba, 2014). We set the learning rate to 1e-04 for BERT and 1e-03 to prototype. We train the model for 20 epochs. The batch size is 20. Then we use the pretrained prototypical OOD classifier to predict unlabeled data. We select the scores in the 5th and last 5th positions after sorting as the upper and lower thresholds. These two thresholds are fixed in each pseudo-labeling process. Those with scores higher than the threshold are classified as prediction, and those with scores lower than the lower threshold are classified as OOD. The remaining samples are still regarded as unlabeled samples and do not participate in training. The coefficients of the three losses are $1.0(\mathcal{L}_{pcl})$, $0.05(\mathcal{L}_{ind})$ and $0.05(\mathcal{L}_{ood})$. For two margin losses, the IND margin value $\mathcal{M}_{IND}$ is 2.8 and the OOD margin value $\mathcal{M}_{OOD}$ is 1.1. During inference, we calculate the

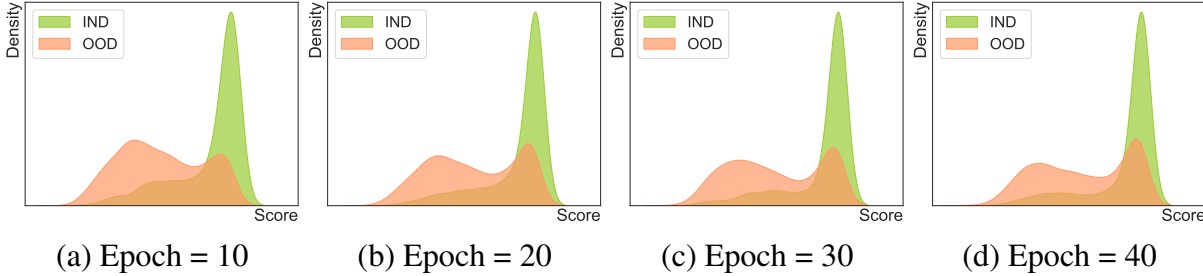

(a) Epoch = 10   (b) Epoch = 20   (c) Epoch = 30   (d) Epoch = 40

Figure 7: Change of score distribution curve of IND and OOD data during pseudo-labeling with $\mathcal{L}_{pcl}$. Since the change trend is the same when using $\mathcal{L}_{pcl}$ and $\mathcal{L}_{pcl} + \mathcal{L}_{ind}$ for self-training, we only show one of them here.

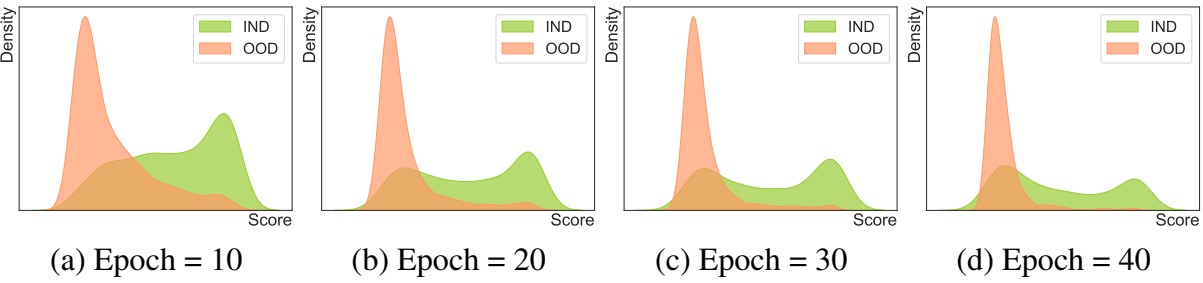

(a) Epoch = 10   (b) Epoch = 20   (c) Epoch = 30   (d) Epoch = 40

Figure 8: Change of score distribution curve of IND and OOD data during pseudo-labeling with $\mathcal{L}_{pcl} + \mathcal{L}_{ood}$.

confidence score of IND data in the validation set, sort it from large to small, and select the score at 75% as the threshold. We conducted a total of 50 epochs pseudo-labeling and pseudo-data training. To avoid randomness, we average results over 3 random runs. The training of prototypical OOD detection lasts about 1.5 minutes and 20 minutes for pseudo-labeling of the second stage both on a single Tesla T4 GPU(16 GB of memory). The average value of the trainable model parameters is 110.18M.

## B  Distribution Changes in Adaptive Pseudo-Labeling

We also show the change of score distribution curves of other self-training methods in Fig 7 and 8. It can be seen that when we remove $\mathcal{L}_{ood}$, a large number of OOD samples will be identified as IND; When we remove $\mathcal{L}_{ind}$, a large number of IND samples will be classified as OOD.

## C  Generality of adaptive prototypical pseudo-labeling

Our adaptive prototypical pseudo-labeling (APP) method can also be combined with other OOD detection methods. In Table 8, we combine APP with SCL+GDA and compare it with our ProtoOOD. The experimental results show that the OOD detection method ProtoOOD proposed by us is signifi-

cantly better than SCL+GDA under the few-shot OOD detection setting. In addition, we use our proposed APP method for self-training on unlabeled data, and find that SCL+GDA also achieve performance improvement, which shows that our APP method has strong generality, and can be compatible with other OOD detection methods well.

| Method | 25% | | | | | | 50% | | | | | | 75% | | | | | |
|---|---|---|---|---|---|---|---|---|---|---|---|---|---|---|---|---|---|---|
| | ALL | | IND | | OOD | | ALL | | IND | | OOD | | ALL | | IND | | OOD | |
| | ACC | F1 | ACC | F1 | Recall | F1 | ACC | F1 | ACC | F1 | Recall | F1 | ACC | F1 | ACC | F1 | Recall | F1 |
| Banking | | | | | | | | | | | | | | | | | | |
| Energy | 66.66 | 48.87 | 60.24 | 47.38 | 68.76 | 77.19 | 60.93 | 57.2 | 61.34 | 56.99 | 60.53 | 65.36 | 58.33 | 61.1 | 59.49 | 61.31 | 55.04 | 49.22 |
| ProtoOOD(ours) | 75.27 | 58.75 | 66.89 | 57.47 | 78.02 | 83.1 | 65.61 | 63.32 | 67.06 | 63.17 | 64.18 | 69.11 | 64.62 | 69.17 | 65.37 | 69.45 | 62.50 | 53.20 |
| APP(ours) | **83.21** | **68.19** | **71.58** | **67.07** | **87.03** | **89.51** | **71.9** | **68.72** | **67.99** | **68.54** | **75.73** | **75.53** | **67.74** | **73.72** | **68.47** | **74.06** | **65.66** | **54.11** |
| Stackoverflow | | | | | | | | | | | | | | | | | | |
| Energy | 45.75 | 39.44 | 57.53 | 36.20 | 41.82 | 55.64 | 49.38 | 48.28 | 53.73 | 47.85 | 45.02 | 52.53 | 49.56 | 52.74 | 51.82 | 53.65 | 42.78 | 39.16 |
| ProtoOOD(ours) | 49.22 | 43.07 | 63.16 | 39.89 | 44.58 | 58.48 | 57.33 | 55.83 | 60.53 | 55.45 | 54.13 | 59.71 | 59.25 | 62.65 | 60.89 | 63.62 | 54.33 | 48.07 |
| APP(ours) | **75.22** | **65.06** | **61.67** | **61.24** | **79.73** | **84.18** | **74.8** | **72.27** | **69.4** | **71.73** | **80.2** | **77.66** | **77.95** | **80.73** | **73.09** | **81.4** | **94.05** | **70.36** |

Table 9: Performance comparison on different IND ratios under 10-shot.