# OpenReview forum: "APP: Adaptive Prototypical Pseudo-Labeling for Few-shot OOD Detection"
_EMNLP/2023/Conference — EMNLP 2023 Findings_

### Official Review · Reviewer_2RaW · 2023-07-29

**Soundness:** 2

**Excitement:**

2: Mediocre: This paper makes marginal contributions (vs non-contemporaneous work), so I would rather not see it in the conference.

**Paper Topic And Main Contributions:**

This paper proposes an adaptive prototypical pseudo-labeling method for few-shot OOD detection, where limited IND data and mixture of IND and OOD data exist. And experiments and analysis show the effectiveness of the proposed method.

**Reasons To Accept:**

1. An adaptive prototypical pseudo-labeling method is developed for few-shot OOD detection.

2. Experiments and analysis show that limited IND data and mixture data can be properly used to enhance OOD detection performance.

**Reasons To Reject:**

1. The problem setting of few-shot OOD detection is new in this work. Previous work such as "Few-shot Out-of-Distribution Detection" and "A Closer Look at Few-Shot Out-of-Distribution Intent Detection" should properly cited and compared.

2. The proposed method a combination of existing methods.

3. Baselines should include previous few-shot OOD detection methods and OOD detector using unlabeled data like "Adversarial self-supervised learning for out-of-domain detection"

**Reproducibility:**

2: Would be hard pressed to reproduce the results. The contribution depends on data that are simply not available outside the author's institution or consortium; not enough details are provided.

**Reviewer Confidence:**

3: Pretty sure, but there's a chance I missed something. Although I have a good feel for this area in general, I did not carefully check the paper's details, e.g., the math, experimental design, or novelty.

---

> ### Author Rebuttal · Authors · 2023-08-29
>
> We appreciate your thorough review and valuable feedback on our manuscript. We are encouraged by your recognition of our contributions. We sincerely apologize for any unclear presentation and hope this response can resolve your concerns.
>
> *Q1. The problem setting of few-shot OOD detection is new in this work. Previous work such as "Few-shot Out-of-Distribution Detection" and "A Closer Look at Few-Shot Out-of-Distribution Intent Detection" should properly cited and compared.*
> A1：Although this paper and the two papers you mentioned are all about the problem of few-shot OOD detection, the setting of this paper is fundamentally different from those methods. **The OOD intent detection in this paper not only involves a small number of few-shot IND samples, but also a large amount of unlabeled mixed data from IND and OOD.** Therefore, we cannot directly compare this paper with the previous articles using their methods. Thank you very much for your attention to the issue of insufficient references in our paper. We will add these references to the paper.
>
> *Q2.The proposed method a combination of existing methods.*
> A2: Instead of copying the previous work, **we creatively apply PCL and self-training to new problems and add targeted methodological innovations like margin loss and adaptive pseudo-labeling process.** We conducted a detailed analysis of the dynamic process of pseudo-labeling in the second stage, and the experimental results demonstrated the effectiveness of our method. **Besides, the few-shot OOD detection with amount of unlabeled data is the first proposed by us, which is a more practical setting than before.**
>
> *Q3.Baselines should include previous few-shot OOD detection methods and OOD detector using unlabeled data like "Adversarial self-supervised learning for out-of-domain detection"*
> A3: About previous few-shot OOD detection methods, we have expressed our opinion in A1. As for the OOD detector using unlabeled data like "Adversarial self-supervised learning for out-of-domain detection". Its motivation is to combine the advantages of unsupervised and supervised methods to extract distinguishable features of OOD intents from unlabeled data. It is different from our research perspective. As we mentioned in the paper, our proposed method is to address a new few-shot scenario, where we face the following problems: **1) how to model IND intents with a small number of IND samples; 2) how to use the preliminary IND intent representation learned in the first stage (which may be inaccurate and unstable) to supplement the diversity loss caused by the small number of IND samples using unlabeled data, avoid noise interference in unlabeled data, and learn the features of OOD samples.** Therefore, the other methods are not suitable for solving this specific setting.

---

### Official Review · Reviewer_tHSS · 2023-08-04

**Typos Grammar Style And Presentation Improvements:** 1. In Figure 3, the legend and other …
**Soundness:** 3

**Excitement:**

3: Ambivalent: It has merits (e.g., it reports state-of-the-art results, the idea is nice), but there are key weaknesses (e.g., it describes incremental work), and it can significantly benefit from another round of revision. However, I won't object to accepting it if my co-reviewers champion it.

**Missing References:**

[1] Ren, Mengye, et al. "Meta-learning for semi-supervised few-shot classification." 6th International Conference on Learning Representations, ICLR 2018. 2018.

**Paper Topic And Main Contributions:**

This paper presents a ProtoOOD framework for out-of-domain intent detection in few-shot scenarios. Additionally, a novel adaptive pseudo-labeling method (APP) is proposed to generate high-quality pseudo out-of-domain (OOD) and in-domain (IND) labels, facilitating OOD detection using limited IND data. Extensive experiments demonstrate the effectiveness of this framework.

**Questions For The Authors:**

1. In the Limitations, the authors point out that the proposed method still has significant gaps compared to full-data OOD detection. In contrast to semi-supervised OOD detection[1], does the APP method have any advantage?
2. Can the effectiveness of $L_{ins}$ and $L_{proto}$ be demonstrated through ablation studying?
3. How is the hyperparameter 0.05 for the final loss $L$ determined at line 240? What is the robustness of the model for different hyperparameter choices?
4. Why does the F1 score of IND improve but the ACC decrease when adding the Adaptive Pseudo-Labeling method to the Stackoverflow dataset, compared to ProtoOOD in Table 2?
5. Where are the pseudo IND samples in Figure 1?


**Reasons To Accept:**

1. The setting in this paper is practical, where there are only a few labeled IND data and massive unlabeled mixed data that may belong to IND or OOD.
2. The motivation is clear and the proposed framework is simple and intuitive.
3. The extensive experimental results demonstrate the effectiveness of the proposed framework to some extent.
4. The paper is well written for readability.


**Reasons To Reject:**

1. The methodology is not novel enough. Similar techniques are commonly applied in representation learning, such as InfoNCE loss, triplet loss, etc.
2. In Table 1 & 2, the latest LLM, e.g., ChatGPT is not compared. Moreover, some results in Table 2 are mistakenly bolded, e.g., ACC for 20-shot IND, recall & F1 for 20-shot OOD.
3. The robustness of coefficients before $L_{ind}$ and $L_{ood}$ is not discussed in the experiments, which I think would have a crucial impact on the generalization of the proposed method.
4. The data set's division lacks clarity, which I think would affect reproducibility.


**Reproducibility:**

3: Could reproduce the results with some difficulty. The settings of parameters are underspecified or subjectively determined; the training/evaluation data are not widely available.

**Reviewer Confidence:**

3: Pretty sure, but there's a chance I missed something. Although I have a good feel for this area in general, I did not carefully check the paper's details, e.g., the math, experimental design, or novelty.

---

> ### Author Rebuttal · Authors · 2023-08-29
>
> We appreciate your thorough review and valuable feedback on our manuscript. We are encouraged by your recognition of the practical setting, clear motivation, simple and intuitive framework, extensive experimental results, and readability of our paper. We sincerely apologize for any unclear presentation and hope this response can resolve your concerns.
>
> *Q1: The methodology is not novel enough. Similar techniques are commonly applied in representation learning, such as InfoNCE loss, triplet loss, etc.*
> A1: Although InfoNCE loss and triplet loss are commonly used techniques in representation learning, our proposed method is significantly different from these two losses. First of all, our method is based on prototype for few-shot OOD detection. In the first stage, we propose to learn IND representation using prototype contrastive learning. In the second stage, we design two prototype-based boundary losses to learn more diverse intent representations during self-training. Our representation learning method is more suitable for OOD detection and few-shot learning scenarios. In addition, our method has achieved very good results in experiments, demonstrating its effectiveness and superiority in practical applications.
>
> *Q2: In Table 1 & 2, the latest LLM, e.g., ChatGPT is not compared. Moreover, some results in Table 2 are mistakenly bolded, e.g., ACC for 20-shot IND, recall & F1 for 20-shot OOD*
> A2: We agree with the issues you mentioned and we need to consider the effectiveness of LLM in OOD detection in our research. We add the experimental results of ChatGPT in zero-shot and few-shot OOD detection, and the data set division is consistent with the previous experiments (Banking-25%). Due to the input length limitation of ChatGPT, we tested OOD detection under 0-shot, 1-shot, 3-shot, and 5-shot settings. The experimental results are as follows.
>
> | Model | ACC-ALL | F1-ALL | ACC-IND | F1-IND | Recall-OOD | F1-OOD |
> | --- | --- | --- | --- | --- | --- | --- |
> | APP（5-shot） | 58.35 | 44.87 | 51.53 | 39.68 | 60.62 | 70.81 |
> | APP (20-shot) | 83.21 | 68.19 | 71.58 | 67.07 | 87.03 | 89.51 |
> | Chatgpt (0-shot) | 51.4 | 45.8 | 77.94 | 45.11 | 42.72 | 58.84 |
> | Chatgpt (1-shot) | 48.96 | 47.11 | 82.46 | 46.72 | 38.26 | 54.45 |
> | Chatgpt (3-shot) | 47.68 | 49.56 | 86.2 | 49.47 | 35.03 | 51.35 |
> | Chatgpt (5-shot) | 44.39 | 47.11 | 89.78 | 47.21 | 29.57 | 45.32 |
>
> We find that ChatGPT's ability to detect OOD samples is quite poor resulting in a very low Recall-OOD (29.57 for 5-shot). Through case analysis, we discover that it tends to classify OOD samples as IND samples. This may be due to a conflict between its vast world knowledge and domain-specific knowledge. **However, the weak OOD indicator of ChatGPT means that it cannot filter out abnormal samples, which is usually the first step in practical applications of the OOD task and also the purpose of OOD task.** Certainly, we also find that its vast world knowledge has brought it excellent IND sample classification capabilities. In our future work, we will attempt to combine the strengths of both models to achieve better overall performance.
>
> Note: We tried various prompts, mainly considering three factors: 1) telling ChatGPT whether it is an intent detector or an OOD intent detector; 2) the order of each part of the prompt; 3) whether to include few-shot samples or intent descriptions. In the end, we chose the optimal prompt as follows:
>
> zero-shot prompt：
> <Task description>
> You are an out-of-domain intent detector, and
> your task is to detect whether the intents of
> users' queries belong to the intents supported
> by the system. If they do, return the
> corresponding intent label, otherwise return
> unknown. The supported intents include:
> [Intent 1], [Intent 2] [Intent N]
> <Response format>
> Please respond to me with the format of
> "Intent: XX" or "Intent: unknown".
> <Utterance for test>
> Please tell me the intent of this text: [Here is
> the utterance for text.]
>
> few-shot prompt:
> <Task description>
> You are an out-of-domain intent detector, and
> your task is to detect whether the intents of
> users' queries belong to the intents supported by
> the system. If they do, return the corresponding
> intent label, otherwise return unknown. The
> supported intents include:
> [Intent 1] ([Example1] [Example 2]...),
> [Intent 2] ([Example1] [Example 2]...), ...
> The text in parentheses is the example of the
> corresponding intent.
> <Response format>
> Please respond to me with the format of
> "Intent: XX" or "Intent: unknown".
> <Utterance for test>
> Please tell me the intent of this text: [Here is the
> utterance for text.]
>
> Q3.The robustness of coefficients before and is not discussed in the experiments, which I think would have a crucial impact on the generalization of the proposed method.
> A3: Thank you for your correction. We verify through experimental results that this parameter value is an optimal value. The parameter requires a value between 0.01 and 0.001 to obtain a stable result. Here are the results of our ablation experiments in the following table:
>
> | parameter（Banking -25% 10-shot） | ACC-ALL | F1-ALL | ACC-IND | F1-IND | Recall-OOD | F1-OOD |
> | --- | --- | --- | --- | --- | --- | --- |
> | 0.03 | 81.10 | 66.06 | 72.11 | 66.21 | 85.0 | 88.79 |
> | 0.05 | 83.21  | 68.19  | 71.58 |  67.07  | 87.03 |  89.51 |
> | 0.07 | 82.83 | 67.09 | 70.89 | 65.97 | 86.70 | 90.23 |
>
> **The results demonstrate that our parameters have a certain degree of robustness and can provide a stable outcome.**
>
> *Q4: The data set's division lacks clarity, which I think would affect reproducibility.*
> A4: we adopted three partition methods, which are 25%, 50%, and 75%, respectively. The experiments were conducted and compared under the same partition for each method. We will add the specific partition details in the appendix. Thanks for your correction.
>
> *Q5：In the Limitations, the authors point out that the proposed method still has significant gaps compared to full-data OOD detection. In contrast to semi-supervised OOD detection[1], does the APP method have any advantage?*
> A5: Thank you for your question. The method [1] you mentioned is a few-shot semi-supervised learning classification algorithm, which extracts features through Prototypical Network and then clusters the features. Because our task involves a large number of OOD samples, which may come from many different intents, we cannot simply transfer the classification method to the OOD detection task. We think this can be a direction for us to explore in future research.
>
> *Q6：Can the effectiveness of and be demonstrated through ablation studying?*
> A6: Our ablation experiments mainly demonstrate the differences between Lpcl, Lpcl+Loos, and APP in terms of changes in IND and OOD scores and the number of correctly labeled samples. We did not include the results of Lpcl+Lins in the ablation experiments, as we have mentioned the reason for this at line 448. The reason is using Lpcl+Lins gets similar results as using only Lpcl, so we only show the results of Cc for brevity.
>
> *Q7:How is the hyperparameter 0.05 for the final loss determined at line 240? What is the robustness of the model for different hyperparameter choices?*
> A7: We have added the results of the ablation experiments and the analysis in the answer of Q3
>
> *Q8. Why does the F1 score of IND improve but the ACC decrease when adding the Adaptive Pseudo-Labeling method to the Stackoverflow dataset, compared to ProtoOOD in Table 2?*
> A8: Thanks for your question. We also notice this experimental result. First, the decrease in IND-ACC and the increase in IND-Precision leads to an increase in IND-F1, as can be seen from the data in the following table. Second, we analyze the reason for this, which may be due to the learning of the representation of IND intent through the few-shot learning in the first stage. After using the adaptive learning algorithm, a large number of OOD samples can be prevented from being misclassified as IND intent, leading to an increase in precision. However, due to the lack of diversity in the features learned through 5-shot, self-learning may introduce noise to the detection of IND intent, leading to a decrease in ACC.
>
> *Q9: Where are the pseudo IND samples in Figure 1?*
> A9: The true IND samples are in deep orange color, while the fake IND samples are in light orange color. Thank you very much for pointing out the issue with the images. We will immediately correct it in the paper.

---

### Official Review · Reviewer_LnTL · 2023-08-05

**Soundness:** 3

**Excitement:**

3: Ambivalent: It has merits (e.g., it reports state-of-the-art results, the idea is nice), but there are key weaknesses (e.g., it describes incremental work), and it can significantly benefit from another round of revision. However, I won't object to accepting it if my co-reviewers champion it.

**Paper Topic And Main Contributions:**

This paper addresses the problem of few-shot out-of-domain (OOD) intent detection in task-oriented dialogue systems. Unlike previous studies that assume plenty of labeled in-domain (IND) intents, this paper focuses on a more practical scenario with limited labeled IND data and massive unlabeled mixed data that may belong to IND or OOD. The paper proposes an adaptive prototypical pseudo-labeling (APP) method, which includes a prototypical OOD detection framework (ProtoOOD) for low-resource OOD detection using limited IND data and an adaptive pseudo-labeling method to produce high-quality pseudo OOD and IND labels from unlabeled mixed data. The experiments and analysis demonstrate the effectiveness of the proposed method for few-shot OOD detection in this challenging setting.

**Reasons To Accept:**

The paper studies an important problem to detect out-of-domain intent. The paper proposes an adaptive prototypical pseudo-labeling (APP) method, which combines a prototypical OOD detection framework (ProtoOOD) with an adaptive pseudo-labeling strategy.  The proposed method achieves promising results and conducted comprehensive results.

**Reasons To Reject:**

1. In few-shot leanring,  in-context examples and prompting strategies are also proven to be effective and play an important role in LLMs. How does the proposed prototypical method compare to those methods?  How does the proposed method compare to LLMs in zero-shot setting or few-shot setting ?
2. The current OOD detection setting has limitations as it only detects out-of-domain instances without specifying the exact intent. A more challenging and practical scenario is to generate new intents in a natural language format. Unlike the existing formulation, a generation framework is not restricted to predefined labels, enabling it to create human-understandable new intents based on given examples.

**Reproducibility:**

3: Could reproduce the results with some difficulty. The settings of parameters are underspecified or subjectively determined; the training/evaluation data are not widely available.

**Reviewer Confidence:**

3: Pretty sure, but there's a chance I missed something. Although I have a good feel for this area in general, I did not carefully check the paper's details, e.g., the math, experimental design, or novelty.

---

> ### Author Rebuttal · Authors · 2023-08-29
>
> We appreciate your thorough review and valuable feedback on our manuscript. We are encouraged by your recognition of our contributions. We sincerely apologize for any unclear presentation and hope this response can resolve your concerns.
>
> *Q1:  In few-shot learning, in-context examples and prompting strategies are also proven to be effective and play an important role in LLMs. How does the proposed prototypical method compare to those methods? How does the proposed method compare to LLMs in zero-shot setting or few-shot setting ?*
> A1: Currently, LLM and the techniques like in-context learning are so popular. We agree with the issues you mentioned and we need to consider the effectiveness of LLM in OOD detection in our research. Therefore, we add the experimental results of ChatGPT in zero-shot and few-shot OOD detection, and the data set division is consistent with the previous experiments (Banking-25%). Due to the input length limitation of ChatGPT, we tested OOD detection under 0-shot, 1-shot, 3-shot, and 5-shot settings. More samples will exceed the input length of ChatGPT. The experimental results are as follows.
>
> | Model | ACC-ALL | F1-ALL | ACC-IND | F1-IND | Recall-OOD | F1-OOD |
> | --- | --- | --- | --- | --- | --- | --- |
> | APP（5-shot） | 58.35 | 44.87 | 51.53 | 39.68 | 60.62 | 70.81 |
> | APP (10-shot) | 83.21 | 68.19 | 71.58 | 67.07 | 87.03 | 89.51 |
> | Chatgpt (0-shot) | 51.4 | 45.8 | 77.94 | 45.11 | 42.72 | 58.84 |
> | Chatgpt (1-shot) | 48.96 | 47.11 | 82.46 | 46.72 | 38.26 | 54.45 |
> | Chatgpt (3-shot) | 47.68 | 49.56 | 86.2 | 49.47 | 35.03 | 51.35 |
> | Chatgpt (5-shot) | 44.39 | 47.11 | 89.78 | 47.21 | 29.57 | 45.32 |
>
> We find that ChatGPT's ability to detect OOD samples is quite poor resulting in a very low Recall-OOD (29.57 for 5-shot). Through case analysis, we discover that it tends to classify OOD samples as IND samples. This may be due to a conflict between its vast world knowledge and domain-specific knowledge. **However, the weak OOD indicator of ChatGPT means that it cannot filter out abnormal samples, which is usually the first step in practical applications of the OOD task and also the purpose of OOD task.** Certainly, we also find that its vast world knowledge has brought it excellent IND sample classification capabilities. In our future work, we will attempt to combine the strengths of both models to achieve better overall performance.
>
> Note: We tried various prompts, mainly considering three factors: 1) telling ChatGPT whether it is an intent detector or an OOD intent detector; 2) the order of each part of the prompt; 3) whether to include few-shot samples or intent descriptions. In the end, we chose the optimal prompt as follows:
>
> zero-shot prompt：
> <Task description>
> You are an out-of-domain intent detector, and
> your task is to detect whether the intents of
> users' queries belong to the intents supported
> by the system. If they do, return the
> corresponding intent label, otherwise return
> unknown. The supported intents include:
> [Intent 1], [Intent 2] [Intent N]
> <Response format>
> Please respond to me with the format of
> "Intent: XX" or "Intent: unknown".
> <Utterance for test>
> Please tell me the intent of this text: [Here is
> the utterance for text.]
>
> few-shot prompt :
> <Task description>
> You are an out-of-domain intent detector, and
> your task is to detect whether the intents of
> users' queries belong to the intents supported by
> the system. If they do, return the corresponding
> intent label, otherwise return unknown. The
> supported intents include:
> [Intent 1] ([Example1] [Example 2]...),
> [Intent 2] ([Example1] [Example 2]...), ...
> The text in parentheses is the example of the
> corresponding intent.
> <Response format>
> Please respond to me with the format of
> "Intent: XX" or "Intent: unknown".
> <Utterance for test>
> Please tell me the intent of this text: [Here is the
> utterance for text.]
>
> *Q2：The current OOD detection setting has limitations as it only detects out-of-domain instances without specifying the exact intent. A more challenging and practical scenario is to generate new intents in a natural language format. Unlike the existing formulation, a generation framework is not restricted to predefined labels, enabling it to create human-understandable new intents based on given examples.*
> A2：As shown in the table above, the generative model (ChatGPT) performs well on the IND classification task, but poorly on OOD samples, which may be due to its lack of domain knowledge. However, the universality of large models makes it difficult to resolve this conflict, so traditional OOD detection methods are difficult to abandon. Inspired by you, we attempted to complete the OOD task by prompting the large model to discover the new intent. Below are our experimental results.
>
> | Model | ACC-ALL | F1-ALL | ACC-IND | F1-IND | Recall-OOD | F1-OOD |
> | --- | --- | --- | --- | --- | --- | --- |
> | APP（5-shot） | 58.35 | 44.87 | 51.53 | 39.68 | 60.62 | 70.81 |
> | APP (10-shot) | 83.21 | 68.19 | 71.58 | 67.07 | 87.03 | 89.51 |
> | Chatgpt (OOD detection 3-shot) | 47.68 | 49.56 | 86.2 | 49.47 | 35.03 | 51.35 |
> | Chatgpt (OOD detection- 5-shot) | 44.39 | 47.11 | 89.78 | 47.21 | 29.57 | 45.32 |
> | Chatgpt (new intent generation-3-shot) | 42.27 | 46.78 | 85.7 | 46.96 | 28.02 | 43.3 |
> | Chatgpt (new intent generation-5-shot) | 38.97 | 43.95 | 85.29 | 44.28 | 23.77 | 37.76 |
>
> From the results, it seems that the framework for generating new intents has a lower ability to detect OOD compared to our APP and ChatGPT. However, I believe the question you raised is a very promising research direction, and we hope to conduct further research in the future.
>
> The used prompt:
> <Task description>
> You are an intent detector, and
> your task is to detect whether the intents of
> users' queries belong to the intents supported
> by the system. If they do, return the
> corresponding intent label, otherwise return
> a new intent label. The supported intents include:
> [Intent 1], [Intent 2] [Intent N]
> <Response format>
> Please respond to me with the format of
> "Intent: XX".
> <Utterance for test>
> Please tell me the intent of this text: [Here is
> the utterance for text.]

---

### Meta-Review · Area_Chair_GJ6C · 2023-09-19

**Recommendation:** 3

**Metareview:**

This paper addresses the problem of few-shot out-of-domain (OOD) intent detection in the specific setting of very few labelled in-domain (IND) samples and a large set of unlabelled (mixed OOD/IND) samples.  Two of the reviewers appreciate that the presented work addresses this practical setting and that the experimental results demonstrate the effectiveness of the proposed prototypical pseudo-labelling method.  The same reviewers also asked for a comparison with LLMs, which was addressed by the authors in their rebuttal by providing results for chatGPT, along with an analysis and an explanation how it was used.  The other reviewer expressed a more critical opinion that the paper lacked novelty and originality, which the authors addressed in the rebuttal by discussing the differences between their approach and the methods mentioned by this reviewer.

---

### Meta-Review · Senior_Area_Chairs · 2023-10-05

**Recommendation:** 3

**Metareview:**

meta review

---

### Decision · Program_Chairs · 2023-10-07

**Decision:**

Accept-Findings

**Comment:**

This paper addresses the problem of few-shot out-of-domain (OOD) intent detection in the specific setting of very few labelled in-domain (IND) samples and a large set of unlabelled (mixed OOD/IND) samples.  Two of the reviewers appreciate that the presented work addresses this practical setting and that the experimental results demonstrate the effectiveness of the proposed prototypical pseudo-labelling method.  The same reviewers also asked for a comparison with LLMs, which was addressed by the authors in their rebuttal by providing results for chatGPT, along with an analysis and an explanation how it was used.  The other reviewer expressed a more critical opinion that the paper lacked novelty and originality, which the authors addressed in the rebuttal by discussing the differences between their approach and the methods mentioned by this reviewer.|meta review